# NMR Metabolite Profiling for the Characterization of Vessalico Garlic Ecotype and Bioactivity against *Xanthomonas campestris* pv. *campestris*

**DOI:** 10.3390/plants13091170

**Published:** 2024-04-23

**Authors:** Valeria Iobbi, Valentina Parisi, Anna Paola Lanteri, Norbert Maggi, Mauro Giacomini, Giuliana Drava, Giovanni Minuto, Andrea Minuto, Nunziatina De Tommasi, Angela Bisio

**Affiliations:** 1Department of Pharmacy, University of Genova, Viale Cembrano 4, 16148 Genova, Italy; valeria.iobbi@edu.unige.it (V.I.); giuliana.drava@unige.it (G.D.); 2Department of Pharmacy, University of Salerno, via Giovanni Paolo II 132, 84084 Salerno, Italy; vparisi@unisa.it; 3CERSAA Centro di Sperimentazione e Assistenza Agricola, Regione Rollo 98, 17031 Albenga, Italy; anna.lanteri@rivlig.camcom.it (A.P.L.); giovanni.minuto@rivlig.camcom.it (G.M.); andrea.minuto@rivlig.camcom.it (A.M.); 4Department of Informatics, Bioengineering, Robotics and System Science, University of Genova, via Opera Pia 13, 16145 Genova, Italy; norbert.maggi@ext.unige.it (N.M.); mauro.giacomini@unige.it (M.G.)

**Keywords:** Vessalico garlic, multivariate data analysis, NMR metabolomics, self-organizing maps, garlic, *Xanthomonas campestris* pv. *campestris*

## Abstract

The Italian garlic ecotype “Vessalico” possesses distinct characteristics compared to its French parent cultivars Messidor and Messidrôme, used for sowing, as well as other ecotypes in neighboring regions. However, due to the lack of a standardized seed supply method and cultivation protocol among farmers in the Vessalico area, a need to identify garlic products that align with the Vessalico ecotype arises. In this study, an NMR-based approach followed by multivariate analysis to analyze the chemical composition of Vessalico garlic sourced from 17 different farms, along with its two French parent cultivars, was employed. Self-organizing maps allowed to identify a homogeneous subset of representative samples of the Vessalico ecotype. Through the OPLS-DA model, the most discriminant metabolites based on values of VIP (Variable Influence on Projections) were selected. Among them, *S*-allylcysteine emerged as a potential marker for distinguishing the Vessalico garlic from the French parent cultivars by NMR screening. Additionally, to promote sustainable agricultural practices, the potential of Vessalico garlic extracts and its main components as agrochemicals against *Xanthomonas campestris* pv. *campestris*, responsible for black rot disease, was explored. The crude extract exhibited a MIC of 125 μg/mL, and allicin demonstrated the highest activity among the tested compounds (MIC value of 31.25 μg/mL).

## 1. Introduction

Garlic (*Allium sativum* L.) varieties, ecotype, and cultivars show a high degree of phenotypic plasticity, dependent on environmental conditions and agricultural practices [1,2,3,4,5]. Morphological diversity in garlic is manifested through variations in bulb size, shape, color, and clove arrangement [6,7]. Along with its morphological features, the concentrations of garlic bioactive compounds [8,9,10,11] can range significantly among different garlic varieties [12,13,14,15,16,17,18], leading to differences in flavor intensity and medicinal properties [19,20].

The Vessalico garlic is cultivated in northwest Italy [21,22]. The cloves of two French cultivars, Messidor and Messidrôme [23], are used every year by the Vessalico farmers for sowing. Our previous study compared Vessalico garlic with the two parent cultivars, and defined Vessalico garlic as an agricultural ecotype [22]. The conditions of growth, harvest, and post-harvest appeared to be more important than the original genotype for the composition of garlic clove sulfur compounds [4,22,24]. Nevertheless, farmers in the Vessalico area do not all rely on the same French producers of Messidor and Messidrôme, and they do not all adopt the same agronomic practices regarding the cultivation, defense, and harvest of the product. The aim of the present study was then to identify which farms produce a garlic product different from the two French cultivars, and which can therefore be designated as true “Vessalico garlic”. NMR metabolite profiling is currently used for the unbiased assessment of changes in the presence and relative abundance of small molecules in response to genetic and/or environmental factors [25,26]. In our study, a harvesting campaign involving the main seventeen farms that produce Vessalico garlic, located in the area of production of this ecotype, was performed, and NMR spectroscopy combined with multivariate data analysis were used to characterize the Vessalico garlic metabolites [27,28,29] and select a representative compound as a potential chemical marker to identify the Vessalico garlic from the two French parent cultivars.

In recent years, the use of natural products in sustainable agriculture has gained considerable attention to reduce reliance on synthetic inputs and promote environmentally friendly farming practices [30,31]. Natural products offer a range of benefits for crop production while minimizing adverse impacts on the ecosystem [32,33]. *Xanthomonas campestris* pv. *campestris* (Pam.) Dowson (*Xcc*) is one of the most widespread members of the *Xanthomonas* group of phytopathogens. This Gram-negative bacterium causes a devastating plant disease known as black rot and it represents a serious problem in agricultural production of *Brassicaceae* plants worldwide [34]. Black rot is a systemic vascular disease and the seedborne infection may kill young plants in the seedbed [34,35,36,37]. At present, the existing methods for *X. campestris* control rely on the use of pathogen-free seeds obtained following elimination of infection arising from seeds; however, no treatment has proven to be entirely foolproof [38]. Previous studies have reported the use of plant natural compounds, essential oils, and extracts active on *X. campestris* pv. *campestris* [39,40,41,42]. Garlic extract has been indicated as a valuable resource for organic agriculture owing to its numerous benefits in pest and disease management, soil enrichment, and plant growth promotion [43]. It has been used also for bactericidal and fungicidal activity [24,44,45]. Moreover, garlic extract is currently recognized as an active substance authorized for all purposes for use in organic farming and included in European regulations [46,47,48,49,50]. Thereby, to find new possibilities for the use of the garlic production waste and residues from the sales of marketed bulbs, Vessalico garlic extract was investigated as a possible agrochemical acting against *X. campestris* pv. *Campestris*.

## 2. Results

### 2.1. ^1^H-NMR Compound Identification

The ^1^H-NMR spectrum of a representative garlic accession (accession 12, Appendix A) is reported in Figure 1, containing as insets the high field and downfield regions (A, B, C, D). The spectral resonances were assigned based on the Chenomx 600 MHz library (CL) and custom library (CCL). A combination of NMR spectra (Appendix A), along with comparison with the published data (Appendix A), were then used to confirm metabolite identification. The spectrum showed signals belonging to carbohydrates, organic acids, amino acids, organosulfur compounds, and other metabolites (Appendix A). The high field region from 0.50 ppm to 3.50 ppm showed a signal arising from aliphatic groups of amino acids (leucine, isoleucine, valine, threonine, alanine, lysine, homoserine, glutamine, glutamate, aspartic acid, asparagine, arginine), organic acids (acetic, pyruvic, succinic, and citric), and other metabolites (acetamide, succinylacetone, riboflavin, choline). The singlets at δ_H_ 2.17 and 2.83 were selected as characteristic of *S*-methyl-L-cysteine and methiin, respectively. The middle field region (δ_H_ 3.50–5.60) showed a signal arising from the carbohydrates (fructose, xylose, α-glucose, and sucrose), sometimes strongly overlapping with the amino acid peaks (cystine, proline, glycine, methionine, serine, pyroglutamate), the organic acids derived from carbohydrates (gluconic and lactic acid), and the methyl moiety of trigonelline. The signals, owing to the anomeric protons of sugars, were clearly visible. The downfield region exhibited the weakest signals, arising from fumaric acid, aromatic resonances of amino acids (phenylalanine), and heterocyclic compounds (histidine, trigonelline). Allyl groups of organosulfur compounds (*S*-allyl-L-cysteine, L-alliin, allicin) were also shown in this region. The following components have been described as characteristic of garlic thermal processing: pyroglutamic acid, acetic acid, and succinic acid, and their presence could be related to sample preparation [51].

### 2.2. Multivariate Data Analysis

The spectral data (Appendix A) matrices of garlic accessions produced by seventeen farms from the same geographical area (Vessalico, Valle Arroscia, Imperia, Italy) as well as from the French cultivars of Messidrôme and Messidor were considered. Exploratory multivariate analysis by Principal Component Analysis (PCA) allowed to visualize the complex data structure in a few dimensions: the first four components explained 91.9% of the data variance. The secondary metabolites were found to be the variables with the maximum loadings. When considering the first two Principal Components, the samples of Vessalico, Messidor, and Messidrôme appeared dispersed and overlapped: three groups of significantly correlated variables were detected (Appendix A). However, the biplot on the second and fourth Principal Components showed a better separation between Vessalico and French samples (Figure 2), with Vessalico samples characterized, for example, by low content of methiin (CCL2) and high content of *S*-methyl-L-cysteine (CCL1), *S*-allyl-L-cysteine (CCL3), and allicin (CCL5). Only the Messidor samples appeared to be similar to the Vessalico accessions. A certain separation among the different production locations and farms was also observed (Appendix A).

Data analysis by means of SOMs was then performed. The number of clusters was assessed based on k-means algorithm joined to the Davies–Bouldin index (DBI) [52]. The index allows to identify the most reliable number of clusters that corresponds to a minimum value of DBI. SOMs related to CL and CCL metabolites reported two clusters (orange and green) which included only Vessalico accessions. The dark blue cluster showed that two French accession samples were mostly distributed toward the top part of the map. Moreover, some Vessalico samples exhibited several replicates in the orange cluster (e.g., 11, 12, 14), suggesting the possibility to find a homogeneous Vessalico product (Appendix A). The PC projection showed a certain separation of Vessalico samples from the French ones (Appendix A). Methiin (CCL2) and L-alliin (CCL4) were the most abundant organosulfur compounds, and gluconic acid (CL24), methionine (CL25), serine (CL26), lactic acid (CL28), and pyroglutamic acid (CL29), followed by fructose (CL27) and sucrose (CL33), were the prevalent compounds among the other metabolites (Appendix A).

Following these findings, the data analysis was repeated, with the CL and CCL metabolites being considered separately, with the aim to better investigate the Vessalico ecotype. This approach could provide more information in the search for a Vessalico biomarker.

SOMs results relative to the CL metabolites showed three clusters containing only Vessalico garlic (pale yellow, yellow, green). No cluster was represented by only French accessions (Appendix A). Nevertheless, the PC projection showed no significative distance of Vessalico accessions from the French ones (Appendix A). No significant differences among the relative content of CL compounds were shown by the U-matrix, unless gluconic acid (CL24), methionine (CL25), serine (CL26), lactic acid (CL28), and pyroglutamic acid (CL29), followed by fructose (CL27) and sucrose (CL33), were the most representative in all the neurons. Trigonelline (CL30) was present in very low amounts (Appendix A). Based on these results, primary metabolites appeared not to be crucial to select a farm producing a homogeneous “Vessalico garlic”.

SOMs related to CCL metabolites showed that the neurons were not specially characterized by the relative high content of certain compounds. Among the seven clusters reported as the best number of clusters (Figure 3A), the map clusterization results allowed to define two clusters of the Vessalico ecotype. The blue and orange clusters (in the left part of the map) were characterized by the sole presence of the selected ecotype with no inclusion of any French samples. The yellow cluster was occupied by the French cultivars, confirming their similarity. The top-right part of the map (green and dark green clusters) was predominantly occupied by Vessalico accessions with three samples of Messidrôme accession (19). Accession 18 (Messidor) was in the purple and light blue clusters on the right part of the map. It exhibited great similarity with respect to various Vessalico accessions (Figure 3B).

The PC projection confirmed these observations, showing a clear separation between Vessalico accessions and French cultivars (Figure 4A,B). The only Vessalico accession included in the same cluster as French cultivars was the number 12 (Figure 3B). All the replicates of this accessions were found in the orange cluster reported in Figure 4C, bordering only with samples from the Vessalico region, and no French accession. These results allowed us to assume that the product of farm 12 was the only homogeneous product of Vessalico garlic. Farm 14 was represented by five samples in the orange cluster with one outlier. Farm 11 was characterized by three samples in the orange cluster and three in the blue one. A relatively higher content of methiin (CCL2) and L-alliin (CCL4) could be attributed to Vessalico accessions of the orange cluster, although no significant differences were observed (Figure 4C and Figure 5).

An Orthogonal Partial Least Squares Discriminant Analysis (OPLS-DA) was applied to a reduced data matrix containing only the typical Vessalico accessions, as obtained from SOM analysis (accessions 11, 12, 14), together with the French accessions (accessions 18, 19, 20). This reduced two-class data matrix contained 18 spectra belonging to each of the two categories (Vessalico and France). An OPLS-DA model with two components was computed and validated, with R^2^(Y) = 0.859 and Q^2^(Y) = 0.764. This model allowed to discriminate between the Vessalico ecotype and the French varieties (Figure 6), showing high selectivity and specificity as confirmed by the misclassification matrix (94% of correct classifications; 89% of correct predictions).

The OPLS-DA method was also used to select the most discriminant metabolites based on values of VIP (Variable Influence on Projections) greater than 1. VIP variables for this set of data are shown in Figure 7.

ANOVA confirmed that all variables having VIP > 1 were significantly different (*p* < 0.001) between the selected Vessalico garlic and French cultivars.

Finally, the model was implemented to predict the class of the remaining 84 samples from Vessalico, not used for the computation of the model. Despite the small number of samples in the training set with respect to the large number of test samples, the model showed good performance, accepting 55% of the Vessalico samples.

### 2.3. S-Allyl-L-Cysteine Quantification

Among the most discriminant variables, sulfur compounds, which are mainly responsible for the garlic aroma, were considered to find a simple way to distinguish between the accession identified as representative of the Vessalico garlic (accession 12) and that identified as representative of the two French parent cultivars. *S*-allyl-L-cysteine was selected to be quantified in the three extracts due to its stability compared to L-alliin and allicin, and because it had an easily identifiable multiplet at 5.81 ppm (Figure 8). The quantification of *S*-allyl-L-cysteine in the garlic extract was carried out through qNMR using a 1D-NOESY sequence. Results showed that the *S*-allyl-L-cysteine content was 135.67 ± 2.18 µg/g in fresh Vessalico garlic cloves, while it was present in negligible amounts in the parent cultivars Messidor and Messidrôme.

### 2.4. Antibacterial Activity

The minimum bacterial concentrations (MICs) of the extract of accession 12 was determined against two strains of *Xanthomonas campestris* pv. *campestris* by using the diluting broth technique. Pure sulfur compounds (S-allyl-L-cysteine, S-methyl-L-cysteine, L-alliin, allicin, and methiin) were also tested. Ampicillin and streptomycin sulphate were included in the test as references for their activity on Gram-negative bacteria [53,54,55,56,57,58]. Allicin proved to be the most active substance against both *X. campestris* pv. *campestris* strains, with a MIC value of 31.25 μg/mL (Table 1). The crude extract was characterized by a MIC value of 125 μg/mL, while S-allyl-L-cysteine, S-methyl-L-cysteine, L-alliin, and methiin showed a MIC of 500 μg/mL against both strains. Ampicillin showed a MIC of 0.25 μg/mL against strain 1 and a MIC value of 0.5 μg/mL against strain 2. Streptomycin sulphate showed a value of MIC of 0.5 μg/mL against strain 1 and a MIC value of 1 μg/mL against strain 2. The growth of *X. campestris* pv. *campestris* was visible in drug-free wells (control of growth) and no growth was observed in wells containing non-inoculated sterile Mueller Hinton Broth (MHB) medium as a blank control.

## 3. Discussion

In this study, an NMR metabolite profiling technique was applied as a first approach to identify which farm, among the main producers of garlic in the Vessalico area, produced a product that could be referred to as true “Vessalico garlic” that differs and can be distinguished from the two French cultivars used for sowing. The main metabolites of garlic include primary metabolites, such as amino acids and carbohydrates, as well as secondary metabolites, such as organosulfur compounds, and more polar compounds of phenolic and steroidal origin, often glycosylated [8,9,11,59]. Organosulfur compounds in intact garlic cloves include about equal amounts of γ-glutamyl-peptides and *S*-alk(en)yl-L-cysteine sulfoxides (ACSOs) [which include (+)-*S*-allyl-L-cysteine sulfoxide (L-alliin), (+)-*S*-(trans-1-propenyl)-L-cysteine sulfoxide (isoalliin), (+)-*S*-methyl-L-cysteine sulfoxide (methiin), and (1*S*,3*R*,5*S*)-3-carboxy-5-methyl-1,4-thiazane-1-oxide (cycloalliin)]. Intermediate compounds in the biosynthesis of ACSOs from γ-glutamyl peptides are *S*-alk(en)yl-cysteines such as (+)-*S*-allyl-L-cysteine (SAC), (+)-*S*-(trans-1-propenyl)-L-cysteine (SPC), and (+)-*S*-methyl-L-cysteine (SMC). Alkyl alkanethiosulfinates such as *S*-allyl cysteine sulfoxide (allyl 2-propenethiosulfinate, allicin) and others are formed from the two main classes of secondary metabolites through enzyme reactions when the raw garlic is cut or crushed [4,9,60]. In our work, we selected five sulfur compounds that could be connected to the various conditions in which garlic bulbs can be found after harvesting and storage [61,62]. Upon crushing the garlic, Allicin abundance has been shown to amount to 60–90% of the total thiosulfinates [11]. The relative content of *S*-alk(en)yl-L-cysteine sulfoxides (L-alliin and methiin) in garlic is known to be affected by several genetic and environmental factors (e.g., climatic conditions, soil composition, irrigation, fertilization, harvest date, etc.) [63]. The concentration of *S*-methyl-L-cysteine and *S*-allyl-L-cysteine is higher in aged garlic [64].

The preliminary findings obtained through explorative analysis by PCA did not highlight a clear separation between the Vessalico garlic and the French varieties. The results obtained by SOM on the profile of the only CL metabolites did not allow the identification of a homogeneous Vessalico product by cluster distribution. However, SOMs performed on the secondary metabolite data enabled the identification of a homogeneous product belonging only to one cluster which was different from the parent cultivars: that product was identified as true “Vessalico garlic”.

When the spectral data were submitted to supervised multivariate analysis by OPLS-DA, a reliable class model was computed, allowing to highlight the high influence of the variables able to discriminate between the Vessalico garlic and the parent French varieties. According to VIP values, eight representative indicators were identified as characteristic of the Vessalico garlic. The indicators included three sulfur compounds, amino acids, and organic acids: *S*-allyl-L-cysteine (SAC), (±)-*S*-allyl-L-cysteine sulfoxide (L-alliin), allyl 2-propenethiosulfinate (allicin), methionine [65], serine [65], lactic acid (characteristic of raw garlic [51]), gluconic acid [66,67], and pyroglutamic acid [51,68], produced by Maillard reactions and characteristic of aged garlic or produced after thermal processing [51]. The largest VIP value was shown by (±)-*S*-allyl-L-cysteine sulfoxide (L-alliin), which is generally considered to be a major factor in determining the quality of garlic [69]. The harvest of garlic is conducted once a year. The garlic is then stored for up to 12 months before being sold or consumed, and the content of sulfur compounds may vary considerably depending on the period and method of storage [24,64]. L-alliin is a thermolabile compound [70] because of its unstable sulfoxide bond. The literature data about changes in the relative amount of L-alliin at different processing stages of cultivation and storage are variable [24,69,71]. Nonetheless, it is described as one of the most degraded compounds due to prolonged storage [62]. Allicin, one the most important biologically active compounds found in crushed or homogenized garlic, is extremely unstable due to the presence of a thiol group [72,73], and its half-life varies depending on the concentration and temperature of the storage solvent [74]. *S*-allyl-L-cysteine is a very stable compound [75], its content increasing during storage [62], as a result of the hydrolysis of γ-glutamyl-*S*-allyl-L-cysteine (GSAC) by the γ-glutamyl transferase enzyme (γ-GTP, EC 2.3.2.2) [75], and unaffected during fermentation and packing steps, similarly to other organosulfur compounds [62]. Thus, *S*-allyl-L-cysteine was selected as a representative metabolite. This metabolite contributes heavily to the health benefits of garlic and it is well-documented for its antioxidant, anti-apoptotic, anti-inflammatory, anti-obesity, cardioprotective, neuroprotective, and hepato-protective properties [76]. *S*-allyl-L-cysteine content in intact garlic is in the range of 19.0–1736.3 μg/g (fresh weight) [75,77,78]. The concentration of *S*-allyl-L-cysteine increases during storage at room temperature (950.0 μg/g) [62], and the content in processed garlic, such as frozen and thawed garlic, pickled garlic, fermented garlic extract, and black garlic is even higher, reaching values of 8021.2 μg/g [78,79,80]. In the present study, the content of *S*-allyl-L-cysteine in the fresh cloves of the garlic accession selected as representative of the Vessalico garlic was 135.67 ± 2.18 µg/g. Since the content of this compound was almost undetectable in the French parent cultivars, *S*-allyl-L-cysteine could be considered a possible chemical marker to distinguish the Vessalico ecotype from the two French accessions.

Although NMR metabolite profiling is currently considered a reliable approach for the assessment of presence and relative abundance of small molecules [27,29,81,82], genetic information is, at present, the only method that provides a final validation with respect to the description of plant ecotypes [83,84,85,86]. Further studies will be needed to explore genetic information, thus providing a more comprehensive characterization of the Vessalico ecotype.

Agroecological pest management [87] is becoming increasingly important as part of a sustainable agriculture vision that incorporates various approaches and areas [30]. The EU bactericides are restricted to a few agents, and non-bactericidal antibiotics are no longer approved [88,89]. In this scenario, the use of plant extracts as antimicrobials is becoming more crucial [90]. Garlic and its bioactive components are well-known for their antimicrobial activity against phytopathogens [45,91], giving rise to the idea of a possible application in biological and integrated pest management. Currently, there is no treatment for the control of black rot caused by *X. campestris*. Consequently, the research of natural products and plant extracts may identify new antimicrobial agents to control this bacterial disease. *X. campestris* pv. *campestris* is the causal agent of black rot of brassicas and is currently a significant problem affecting a large group of horticultural crops grown in the open field. As previously mentioned, its primary spread occurs through the seed, but there are currently no tanning agents available to address this type of problem. Furthermore, in the field, it used to be contained with the use of copper [92] which is undergoing strong regulation [93]. Previously, the use of copper was coupled with the use of dithiocarbamates [94], whose usage is currently no longer allowed in the EU. Some brassicaceous species are also included in the “baby leaf” category for which the use of synthetic products presents strong limitations due to the application of very low residual limits. “Baby leaves” are also one of the crops of choice in “vertical farming”, an emerging growing practice [95]. This type of cultivation involves the exploitation of a confined environment where irrigation water is continuously reused. The environment, therefore, is particularly conducive to bacterial diseases and not very suitable for the application of copper-based compounds often characterized by a marked phytotoxic effect. The present study investigated the antimicrobial activity of garlic extract against *X. campestris* pv. *campestris.* Other plant extracts have been tested against *X. campestris* pv. *campestris*, with MIC values ranging from 0.15 mg/mL to 1.25 mg/mL [96,97,98]. In this study, the MIC of garlic extracts against *X. campestris* pv. *campestris* was 125 μg/mL, thus showing a good activity compared to other extracts discussed in the literature. Allicin is recognized as the main compound responsible for the antimicrobial activity of garlic [99,100,101], and its activity against *X. campestris* pv. *malvacearum* [102] and *X. campestris* pv. *campestris* [45] (Zone of Inhibition Test) has been explored. The MIC of allicin against *X. campestris* pv. *campestris* has been shown to be lower than that of commercial copper-based products against *X. campestris* [103,104] and lower than that of other natural products against *X. campestris* pv. *vesicatoria* (geraniol, MIC = 250 μg/mL; thymol, MIC = 125 μg/mL; o-vanillin, MIC = 250 μg/mL) [105].

## 4. Materials and Methods

### 4.1. General Experimental Procedures

Ultra-pure water and all organic solvents of analytical grade were purchased from ROMIL-UpSTM (Waterbeach, Cambridge, UK). Deuterium oxide (D2O, 99.90% D), CD_3_OD (99.95% D), and 3-(trimethylsilyl) propionic-2,2,3,3-d 4 acid sodium salt (TSP) were purchased from the Sigma-Aldrich chemical company (Sigma-Aldrich, Milano, Italy). The standard compounds chosen as representatives of their main class were *S*-allyl-L-cysteine, *S*-methyl-L-cysteine, and (±)-L-alliin ≥ 90%, all of which were purchased from Merck KGaA (Darmstadt, Germany), allicin, obtained from MedChemExpress (Monmouth Junction, NJ, USA), and methiin, purchased from Abcam (Cambridge, UK).

### 4.2. Plant Material

Seventeen accessions of *Allium sativum* ecotypes (“Vessalico garlic”), grown under field conditions in numerous farms located in six different areas of Valle Arroscia (Imperia, Italy), along with three commercial accessions of the French cultivars Messidrôme and Messidor (one of Messidrôme and two of Messidor, respectively, representing all commercial sources from which farmers acquire their supply) were collected and authenticated based on clove morphology (Appendix A). The identification of all the garlic accessions was performed by Dr. Andrea Minuto [106,107]. The vouchers of all the accessions were deposited at CERSAA (Albenga, Italy).

### 4.3. Sample Collection and Preparation

Harvesting took place in June 2023, when the garlic reached full ripeness, indicated by the falling of the neck or the drying of leaves. The harvested garlic was then stored at cellar temperature. The study was conducted during late autumn, replicating the conditions in which commercial garlic is typically sold throughout the autumn and winter season. A random selection of clove samples was conducted, and each individual undamaged clove was meticulously peeled, frozen, and lyophilized in a freeze-dryer (Super Modulyo, Edwards, UK) for 48 h. Three biological replicates for each accession were used, with a total of 60 samples. All samples were sealed in plastic bags and stored dry in the dark until analysis. The dried material (about 35 g per sample) was then ground.

The powder samples were prepared following the method reported by Tajidin et al. [108]. Briefly, 50 mg of each dried sample was extracted by vortexing (30 s) with 1.0 mL of CD_3_OD (0.5 mL, 99.95%) and KH_2_PO_4_ (0.5 mL) buffer in D_2_O (pH 6.0) containing 0.1% of 3-(trimethylsilyl) propionic-2,2,3,3-d_4_ acid sodium salt (TSP) and then sonicated (30 min) (Branson 2510E-MTH, Bransonic^®^, Milano, Italy) at room temperature. The clear deuterated supernatant obtained after centrifuging (D3024 Microcentrifuge, Scilogex, Rocky Hill, CT, USA) at 13 rpm for 10 min was transferred into NMR tubes. The extracts for the NMR quantification and antimicrobial assays were prepared using 85 g of peeled and crushed cloves with 300 mL of CH_3_OH ≥ 99.9%/H_2_O 1:1 50:50 at 25 °C, then sonicated (VWR USC200TH, VWR International, Leuven, Belgium) at a fixed frequency of 37 KHz for 30 min at room temperature. The supernatant obtained was filtered and evaporated using a rotary evaporator. All the procedures were carried out in a timely manner to avoid degradation of compounds [73,109]. The extract powders were then stored in a laboratory freezer (−20 °C).

### 4.4. NMR Spectroscopy and Processing

NMR data were acquired on a Bruker Ascend™ 600 NMR spectrometer (Bruker BioSpin GmBH, Rheinstetten, Germany) equipped with a Bruker 5 mm TCI CryoProbe at 300 K, operating at 600 MHz, with the temperature maintained at 27 °C, and H_2_O-d_2_ was used as an internal lock. Each ^1^H NMR spectrum consisted of 64 scans, 2.05 s acquisition time, a relaxation delay (RD) of 4 s, mixing time of 0.01 s, and a spectral width of 13.33 ppm (corresponding to 8000 Hz). A pre-saturation sequence (NOESY-presat sequence, Bruker: noesygppr1d) was used to suppress the residual signal of water [110,111,112]. A Chenomx 600 MHz custom library (CCL) (Chenomx NMR Suite 8.6, Chenomx Inc., Edmonton, AB, 252 Canada) was set up by means of pure secondary metabolites obtained from commercial sources (Sigma-Aldrich, Milano, Italy). The Chenomx Compound Builder tool was used. The CCL included five standard compounds: *S*-methyl-L-cysteine (SMC) (CCL1), *S*-methyl-L-cysteine sulfoxide (methiin) (CCL2), *S*-allyl-L-cysteine (SAC) (CCL3), (±)-*S*-allyl-L-cysteine sulfoxide (L-alliin) (CCL4), and allyl 2-propenethiosulfinate (allicin) (CCL5). Additionally, 36 metabolites from the Chenomx 600 MHz library, selected based on literature data [28,29,51,113,114], were used (Appendix A). Each ^1^H NMR spectrum was acquired using the 1D. All spectra were acquired in duplicates. The metabolites were identified based on the comparison of their ^1^H NMR spectra to those of the reference compounds in both the custom and the 600 MHz version libraries (MSI level of identification according to Sumner et al.) [115].

### 4.5. NMR Data Analysis

NMR data were acquired on a Bruker. NMR spectra analysis and metabolite quantification were then performed by using the online server NMRProcFlow (INRA UMR 1332 BFP, Bordeaux Metabolomics Facility, Villenave d’Ornon, France) [116] following the method reported by Grimaldi et al. [117]. Briefly, corrections of phasing and baseline were performed manually for all spectra using TOPSPIN version 3.2. All spectra were calibrated by using the internal standard at 0 ppm. Spectral area integration was achieved by variable sized bucketing using the online server NMRProcFlow. Buckets with a signal-to-noise ratio above 3 were selected for further analysis. The residual solvent regions of water (δ_H_ 4.65–4.75) were removed (Appendix A). The data matrices generated by NMRProcFlow, one of five buckets (CCL compounds) and one of 36 buckets (CL compounds), were then subjected to multivariate analysis. The metabolite identification was assessed by comparison of their ^1^H NMR spectra to those of the Chenomx 600 MHz libraries.

### 4.6. Multivariate Data Analysis

Exploratory data analysis and an ANOVA were performed using the Systat software for Windows Version 13 (Systat Software Inc., Chicago, IL, USA). A Principal Component Analysis (PCA) was conducted on the spectral data after Pareto scaling.

Self-Organizing Maps (SOMs) were employed as an unsupervised model, utilizing Matlab R2022a (MathWorks, Inc., Natick, MA, USA) and SOM Toolbox 2.1 [118]. SOMs are characterized by their ability to organize and process information in a network-like structure. To prepare the data for analysis, a series of pre-processing steps were undertaken. First, the dataset underwent a log transformation, following the approach outlined by van den Berg et al. [119]. This transformation was necessary to mitigate the dominance of variables with higher ranges, as they could disproportionately influence the distances within the map. Additionally, a variance-based normalization technique was applied to ensure balanced representation of the variables. The SOM training process consisted of two distinct phases: the rough phase and the refinement phase. In the rough phase, the SOM was trained with a larger radius and learning rate, an approach which facilitated a more extensive exploration of the data space. This phase also took into consideration the influence of the most distant nodes, enabling a comprehensive representation of the dataset. Following the rough phase, the refinement phase commenced, employing a smaller radius and learning rate to fine-tune the SOM. This phase allowed for localized adjustments, leading to convergence towards a final map representation. Upon completion of the training process, the U-matrix was generated to visually depict the distances between neighboring map units. The U-matrix facilitated cluster identification, with uniform areas indicating distinct clusters and higher values highlighting cluster boundaries. In addition to the U-matrix, other maps were generated to represent the component plan, focusing on single compounds. Highly correlated variables exhibited similar map patterns, enabling insights into the interrelationships among the variables. Hits, defined as the number of times a map unit responded to inputs, were associated with specific units within the map. Hits served as an indicator of the amount of input information collected by each neuron, providing valuable information about the data distribution. The use of SOMs in this study established the framework for subsequent data analysis and interpretation. The described steps ensured proper data pre-processing, effective training, and visualization of the resulting SOM representation, setting the stage for a comprehensive understanding of the underlying patterns within the dataset.

OPLS-DA [120] was used for discriminating the representative samples of the Vessalico ecotype from the French cultivars, using a MATLAB toolbox called PLS_toolbox by Eigenvector. The OPLS-DA model was validated and used to predict the class of test samples (i.e., Vessalico accessions not used for computing the OPLS-DA model). The quality of the model was evaluated in terms of R^2^ and Q^2^ and by means of misclassification matrices [121]. The most characterizing metabolites were selected on the basis of the Variable Influence on Projections (VIP values) of OPLS-DA.

### 4.7. S-Allyl-L-Cysteine Quantification

Among the characterizing metabolites, *S*-allyl-L-cysteine (CCL3) was selected as a representative metabolite, owing to the presence of an isolated doublet at 5.81 ppm, and then quantified in the Vessalico garlic (accession 12), Messidor, and Messidrôme extracts. A calibration curve of SAC was made in a concentration range of 10–500 µg/mL. The linearity of the instrumental response in the analyzed concentration range was confirmed, as inferred by the following fitting curve parameters: y = 787,835, x − 6866.9, R^2^ = 0.9994. The Limit of Detection (LOD) and the Limit of Quantification (LOQ) were determined by serially diluting *S*-allyl-L-cysteine. The analysis was performed until the results of signal-to-noise ratio (S/N) reached the values of 3:1 and 10:1. The obtained values of the LOD and LOQ were 2.0 µg/mL and 8.0 µg/mL, respectively. All the data needed were exported into a spreadsheet workbook using the “qHNMR” template.

### 4.8. Antibacterial Activity

Two strains of *Xanthomonas campestris* pv *campestris* obtained from the microbial collection of CeRSAA were employed in this study. The strains were previously isolated from different symptomatic plant hosts (*Brassica oleracea* and *Eruca vesicaria*), identified by molecular sequencing, and characterized according to the pathogenicity test (Koch postulates confirmation) [122,123]. Sterile stock solutions in 80% dimethyl sulfoxide (DMSO) (Sigma Aldrich, St. Louis, Missouri, USA) of the extract and pure compounds (*S*-allyl-L-cysteine, *S*-methyl-L-cysteine, (±)-L-alliin, allicin, and methiin) (20 mg/mL) were prepared and stored at −20 °C. Dilutions 1:10 of the six stock solutions were obtained using Mueller Hinton Broth (Merck-Millipore, Burglinton, MA, USA). The concentration obtained (2000 μg/mL) was, on two occasions, the highest test concentration, and 100 μL of each solution was transferred in a well of the first column. Sterile stock solutions in 80% dimethyl sulfoxide (DMSO) (Sigma Aldrich, St. Louis, MO, USA) of ampicillin (Sigma-Aldrich, Milano, Italy) and streptomycin sulphate (VWR Life Science, Radnor, PA, USA) (0.64 mg/mL) were prepared and stored at −20 °C. Dilutions 1:10 of the two antibiotics were obtained using Mueller Hinton Broth. The concentration obtained (64 μg/mL) was, on two occasions, the highest test concentration, and 100 µL of each solution was transferred in a well of the first column. One-day-old bacteria cultures were diluted in Buffered Peptone Water (VWR Life Science, Radnor, PA, USA) to obtain a bacterial suspension acclimated to 0.5 on the McFarland scale. Microbial inoculums were then diluted to 1/150 in Mueller Hinton Broth (Merck-Millipore, Burglinton, MA, USA) to obtain a final concentration of approximately 5 × 105 cells/mL. The MICs of extracts and pure compounds of *S*-allyl-L-cysteine, *S*-methyl-L-cysteine, (±)-L-alliin, allicin, and methiin were determined by following the microdilution procedure [124] reported by the Clinical and Laboratory Standards Institute [125] using Mueller Hinton Broth as the test medium. Briefly, 50 µL of inoculum obtained as described above was added to equivalent volumes of various concentrations of extracts and pure compounds of *S*-allyl-L-cysteine, *S*-methyl-L-cysteine, (±)-L-alliin, allicin, and methiin, distributed across a 96-well microplate, and prepared from two-fold serial dilutions ranging from 0.977 µg/mL to 1000 µg/mL. Simultaneously, the inoculum was added to equivalent volumes of ampicillin and streptomycin sulphate distributed across a 96-well microplate and prepared from two-fold serial dilutions ranging from 0.031 µg/mL to 32 µg/mL. The activity of DMSO as a negative control was tested in the last row and it ranged from 0.039 µL/m to 40 µL/mL. The last line contained five drug-free wells control of growth, and three wells containing non-inoculated sterile Mueller Hinton Broth (MHB) medium as a blank control. To avoid degradation of the compounds [73,109], all procedures were executed in a timely manner, and the microplate and solutions were kept on ice during all the working procedures until the following incubation step. After 24 h of incubation in dark conditions at 35 °C, the lowest concentration of compounds preventing visible growth was recorded as the MIC. All MICs were obtained in triplicates.

## 5. Conclusions

In conclusion, the comparative characterization of the Vessalico garlic, along with the French cultivars Messidor and Messidrôme, offers a unique opportunity to explore the intricate relationships between garlic ecotypes and their distinct regional adaptations. The findings of this study hold significant implications for agricultural practices, culinary traditions, and the preservation of cultural heritage, ultimately guiding future conservation efforts and sustainable cultivation practices for these valuable garlic varieties. The NMR metabolomic study, followed by multivariate data analysis, allowed to define the secondary metabolites more related to the area and to the methods of cultivation and harvesting. Moreover, accession 12 was identified as the only product of Vessalico different from the two French parent cultivars. Among the secondary metabolites, *S*-allyl-L-cysteine could be considered as the biomarker to identify the Vessalico garlic among the other French parent cultivars. Future research to obtain genetic data, thereby offering a more comprehensive characterization of the selected ecotype, will be performed.

Although the antimicrobial activity of garlic extracts and allicin is widely documented in the literature [45,126,127], no study on the evaluation of the MICs of pure compounds of *S*-allyl-L-cysteine, S-methyl-L-cysteine, (±)-L-alliin, allicin, and methiin against *X. campestris* pv. *campestris* through the dilution broth technique has been conducted so far. The formation of allicin, the most active substance against *X. campestris* pv. *campestris*, when alliinase cleaves alliin after the cell breaks, is the basis of its specific role in plant defense mechanisms [128]. Given the impossibility of applying antibiotics and synthetic products, as well as the progressive reduction of the legal limits allowed for copper-based products and considering the low environmental impact of plant extracts and pure compounds, the potential use of garlic extracts as well as allicin (in appropriate formulations to avoid its degradation) in the control of infections caused by *X. campestris* pv. *campestris* is currently of great interest and relevance. Our results suggest that garlic extracts could be considered to control the bacterial disease caused by *X. campestris* pv. *campestris*. Nonetheless, further studies in vivo on formulation and protection strategies are needed for use in conventional and organic agriculture.

## Figures and Tables

**Figure 1 plants-13-01170-f001:**
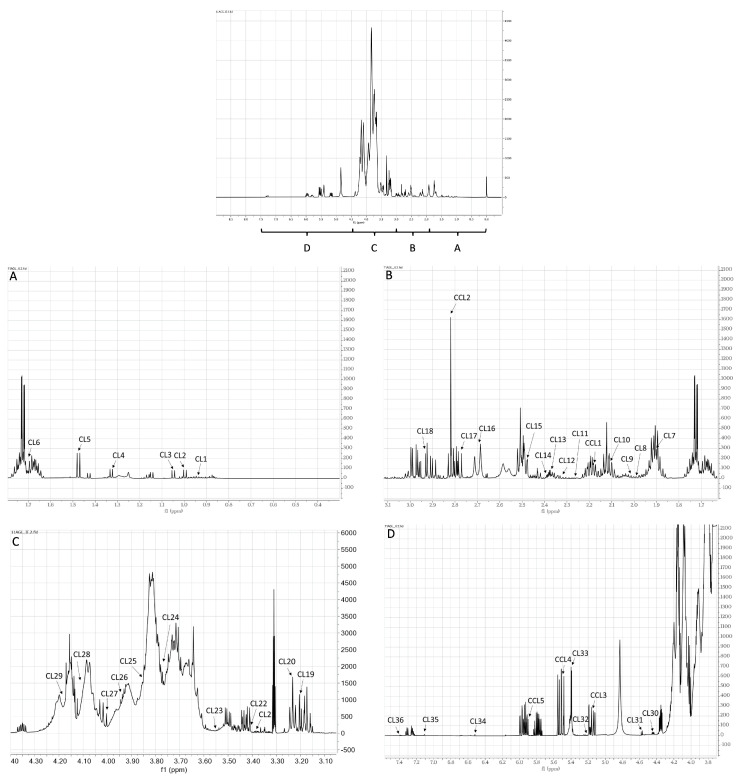
Representative ^1^H-NMR spectrum of the Vessalico garlic (accession 12, Appendix A) in CD_3_OD-KH_2_PO_4_ in D_2_O at pH 6.0, 600 MHz. The spectrum was scaled to internal 1 mM deuterated sodium 3-(trimethylsilyl)-1-propionic acid (TSP), assumed to resonate at 0.00 ppm. The region δ_H_ 0.0–7.5 was expanded in (**A**), (**B**), (**C**) and (**D**), respectively. Identified resonances are labeled according to Appendix A (Appendix A): CCL1: *S*-methyl-L-cysteine, CCL2: methiin, CCL3: *S*-allyl-L-cysteine, CCL4: L-alliin, CCL5: allicin, CL1: leucine, CL2: isoleucine, CL3: valine, CL4: threonine, CL6: alanine, CL5: lysine, CL7: acetic acid, CL8: acetamide, CL9: homoserine, CL10: glutamine, CL11: succinylacetone, CL12: glutamic acid, CL13: pyruvic acid, CL14: succinic acid, CL15: riboflavin, CL16: citric acid, CL17: aspartic acid, CL18: asparagine, CL19: choline, CL20: arginine, CL21: cystine, CL22: proline, CL23: glycine, CL24: gluconic acid, CL25: methionine, CL26: serine, CL27: fructose; CL28: lactic acid, CL29: pyroglutamic acid, CL30: trigonelline, CL31: xylose, CL32: α-glucose, CL33: sucrose, CL34: fumaric acid, CL35: histidine, CL36: phenylalanine.

**Figure 2 plants-13-01170-f002:**
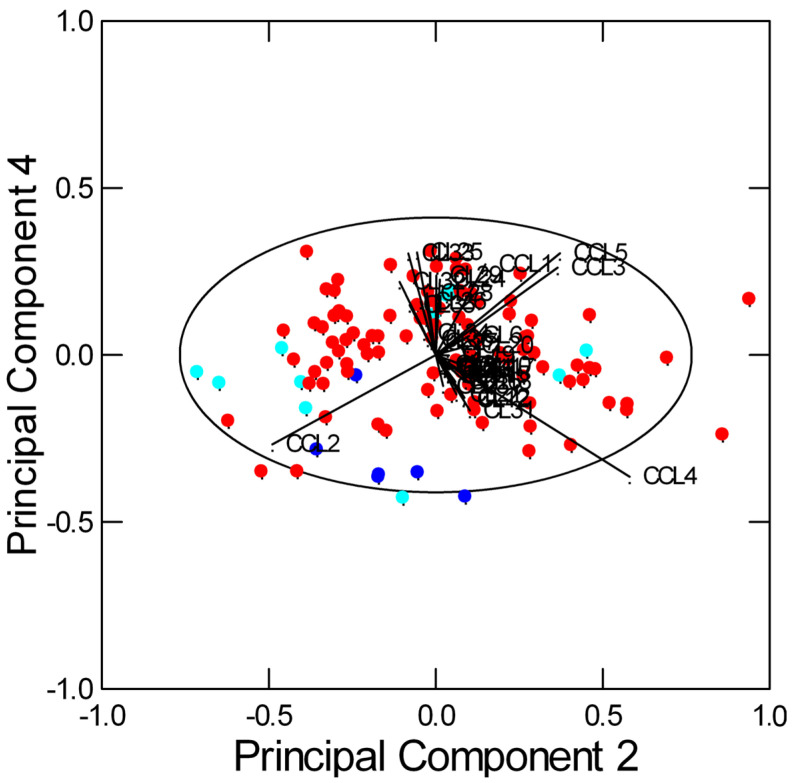
Results of PCA. The biplot shows the scores of the 120 spectra and the loadings of the 41 variables (5 CCL and 36 CL metabolites, Chenomx 600 MHz library and custom library metabolites) on Principal Components 2 and 4 (explaining 27.8% and 8.0% of the total variance, respectively): ● Vessalico; ● Messidor; ● Messidrôme. The ellipse represents the 95% confidence interval.

**Figure 3 plants-13-01170-f003:**
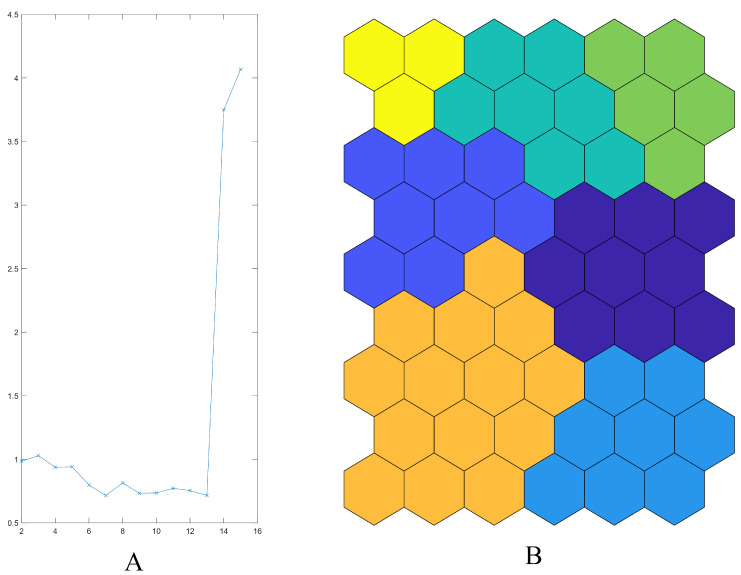
SOM: map clusterization. Five CCL variables (Chenomx 600 MHz custom library metabolites). (**A**) Davies–Bouldine index progression: minimum value fits the best number of clusters; (**B**) Seven clusters.

**Figure 4 plants-13-01170-f004:**
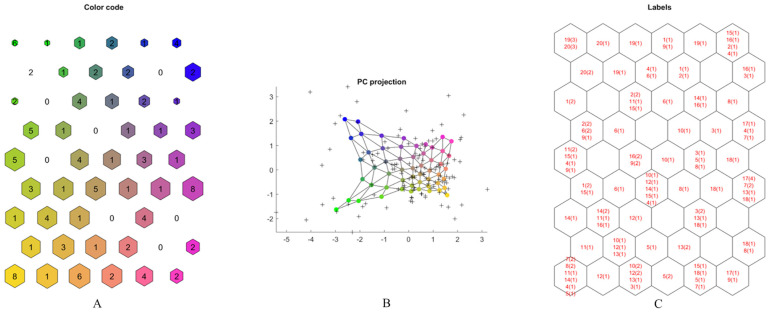
SOM: graphical representation of five CCL variables (Chenomx 600 MHz custom library metabolites) on map. (**A**) SOM output map with color code association. Similar colors have similar characteristics; numbers correspond to hit numbers. The dimensions of the hexagons are related to the distance between neurons (the greater the size, the greater the distance). (**B**) Principal component projection of the map; (**C**) labelled SOM output map, for each neuron the corresponding accession number (Appendix A) and number of replicates (in parentheses) are shown.

**Figure 5 plants-13-01170-f005:**
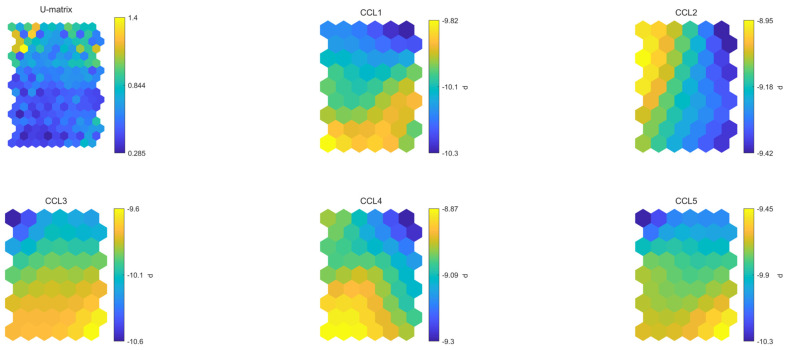
SOM: U-matrix and maps for each CCL variable (Chenomx 600 MHz custom library metabolites). Similar color gradations indicate highly correlated variables. CCL1: *S*-methyl-L-cysteine, CCL2: methiin, CCL3: *S*-allyl-L-cysteine, CCL4: L-alliin, CCL5: allicin.

**Figure 6 plants-13-01170-f006:**
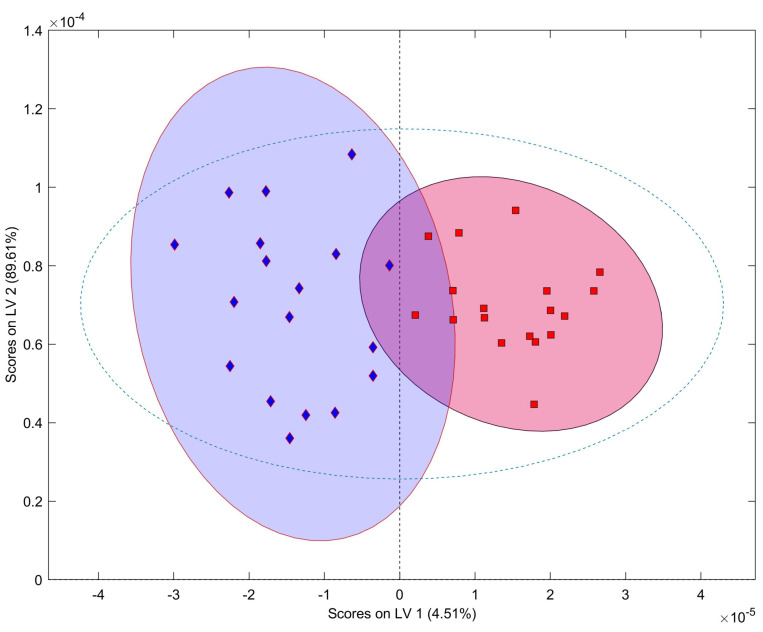
OPLS-DA score plot with confidence ellipses for the two classes: Vessalico (red square) and France (blue diamond).

**Figure 7 plants-13-01170-f007:**
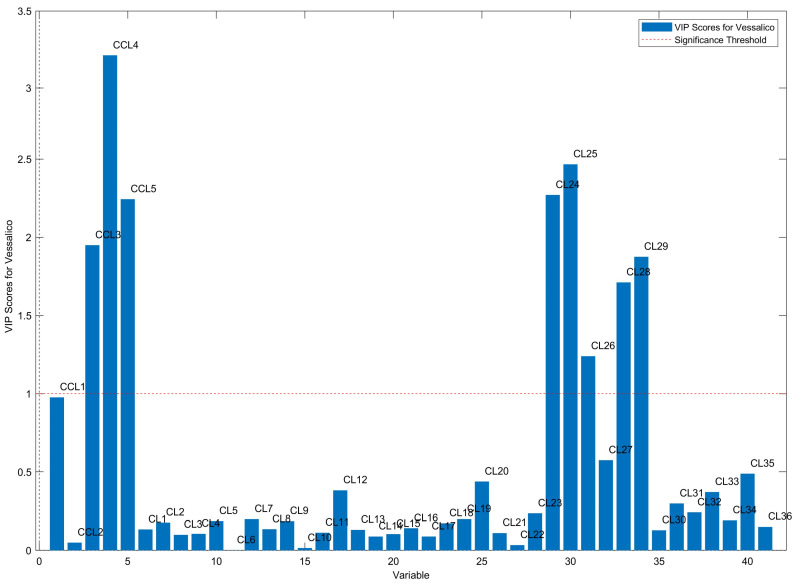
VIP (Variable Influence on Projections) values: the most discriminant variables (VIP > 1) are CCL3 (*S*-allyl-L-cysteine), CCL4 (L-alliin), CCL5 (allicin), CL24 (gluconic acid), CL25 (methionine), CL26 (serine), CL28 (lactic acid), and CL29 (pyroglutamic acid).

**Figure 8 plants-13-01170-f008:**
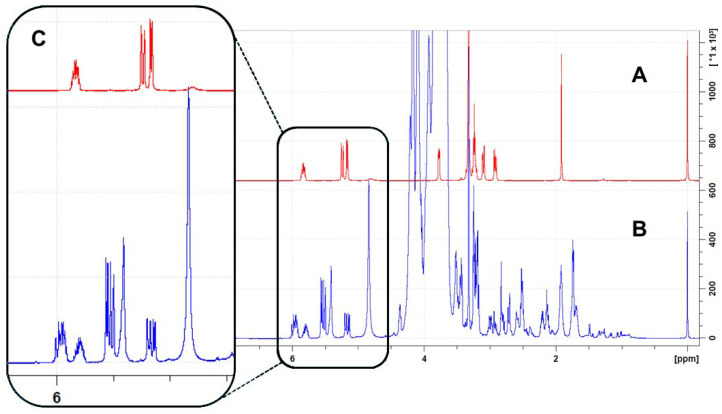
Spectra (^1^H NMR) of *S*-allyl-L-cysteine (**A**), and accession 12 extract (**B**). Zoom of spectra from δ_H_ 4.70 ppm to 6.30 ppm (**C**). The isolated doublet at 5.18 ppm (in the black square) of *S*-allyl-L-cysteine was used for the quantification.

**Table 1 plants-13-01170-t001:** Evaluation of the MICs of extract, pure compounds, and antibiotics against *Xanthomonas campestris* pv. *campestris*.

Treatment	MIC			
	(μg/mL)		μM	
	Strain 1	Strain 2	Strain 1	Strain 2
*S*-allyl-L-cysteine	500	500	3101.3	3101.3
*S*-methyl-L-cysteine	500	500	3698.77	3698.77
L-alliin	500	500	2821.35	2821.35
Methiin	500	500	3307.10	3307.10
Allicin	31.25	31.25	192.57	192.57
Crude extract	125	125	-	-
Ampicillin	0.25	0.5	0.72	1.43
Streptomycin sulphate	0.5	1	0.34	0.69

## Data Availability

The data presented in this study are available in the article or in the Appendix A.

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
