# Peer review of "NMR Metabolite Profiling for the Characterization of Vessalico Garlic Ecotype and Bioactivity against *Xanthomonas campestris* pv. *campestris"

_plants, 2024, doi:10.3390/plants13091170_

Round 1
Reviewer 1 Report
Comments and Suggestions for Authors
The article is probably of more economic interest to growers, if it turns out that a new variety with special and enhanced characteristics can be produced, than purely scientific.
Although the identification of metabolites by NMR is correct as well as the statistical analysis is appropriate, nevertheless several questions were raised about the conclusions.
Initially, the discussion is verbose and not comprehensive. The first paragraph, in particular, can be removed without altering the meaning of the discussion at all.
According to lines 241-242 and 244-245, only samples from one farm appeared to have the characteristics of the Vessalico variety because they were all included in the same cluster. What about the two parent varieties? Are they in the same cluster? If not, how can they be distinguished from Vessalico?
Table 1 gives the MIC values. Of all the sulphur compounds mentioned in the article, only one was tested for its concentration in the extract and that is one which has practically no activity. Why was the content of allicin, which is the only active substance, not also measured in order to check whether the activity of the extract is exclusively due to allicin or whether there is a synergistic effect?
According to lines 400-403, S-allyl-L-cysteine may be considered as the marker because its concentration was 135.67 ± 2.18 µg/g. I honestly do not understand how the authors came to this conclusion. First of all, a single metabolite cannot determine the difference between similar (identical) natural products.
Because of all this, I cannot recommend this article for publication unless the article is rewritten adding more data to prove that there are indeed metabolites that differentiate this particular variety.
Author Response
The article is probably of more economic interest to growers, if it turns out that a new variety with special and enhanced characteristics can be produced, than purely scientific. Although the identification of metabolites by NMR is correct as well as the statistical analysis is appropriate, nevertheless several questions were raised about the conclusions.
Initially, the discussion is verbose and not comprehensive. The first paragraph, in particular, can be removed without altering the meaning of the discussion at all.
We thank the reviewer for the suggestion. The first paragraph of the discussion has been removed following the reviewer suggestion.
According to lines 241-242 and 244-245, only samples from one farm appeared to have the characteristics of the Vessalico variety because they were all included in the same cluster. What about the two parent varieties? Are they in the same cluster? If not, how can they be distinguished from Vessalico?
We appreciate the reviewer's comment, that allowed us to better clarify our results.
Through SOM analysis based on secondary metabolites, seven clusters were identified. The resulting map clusterization showed two distinct clusters representing the Vessalico ecotype. Specifically, the blue and orange clusters included accessions from Vessalico geographical area, with no inclusion of any French samples. This observation was further supported by PC projection, which clearly depicted a separation between Vessalico accessions and French cultivars. The yellow cluster predominantly consisted of French cultivars. Following the reviewer’s suggestion, the results have been modified (Lines 176-177 and 192).
Additionally, the OPLS-DA method (Figure 6), applied to the more homogeneous Vessalico accessions (11, 12, and 14), belonging to the blue and orange clusters, vs the French accessions, confirmed SOM results, highlighting the difference between Vessalico and French accessions.
Table 1 gives the MIC values. Of all the sulphur compounds mentioned in the article, only one was tested for its concentration in the extract and that is one which has practically no activity. Why was the content of allicin, which is the only active substance, not also measured in order to check whether the activity of the extract is exclusively due to allicin or whether there is a synergistic effect?
We thank the reviewer for the comment, that helped us in improving our manuscript. Our work had two main purposes: to find a fairly simple way to characterize the Vessalico garlic and to identify a possible route of use of ground Vessalico garlic extracts as antimicrobials in organic farming for the development of a local sustainable agriculture. Thus, our intent was not to associate a potential method for identifying Vessalico garlic with the biological activity of garlic extracts.
We tested garlic extract and some pure sulfur compounds against X. campestris pv. campestris. Among them, allicin is recognized in the literature as the main responsible for the antimicrobial activity of garlic extracts. It is well known that interactions among metabolites within the crude extract of a plant can give rise to different and opposite results including antagonism, synergism, additivity action, as well as indifference. The possibility of interactions between different compounds justifies the need for a comprehensive microbiological study of a plant extract. This study was beyond what we planned. We used pure sulfur compounds as comparison with the extract MIC value.
Following the reviewer’s suggestion, the introduction, the results, and the conclusions have been modified accordingly (lines 78-79, 258-259, 579-582, respectively)
According to lines 400-403, S-allyl-L-cysteine may be considered as the marker because its concentration was 135.67 ± 2.18 µg/g. I honestly do not understand how the authors came to this conclusion. First of all, a single metabolite cannot determine the difference between similar (identical) natural products.
We regret not having expressed our thought clearly. S-allyl-L-cysteine was selected as representative metabolite for several reasons. Relying on VIP values, three sulfur compounds (i.e. alliin, allicin and S-allyl-L-cysteine) were among the the most discriminant variables to distinguish Vessalico garlic from the French cultivars.
Among the three sulfur compounds, S-allyl-L-cysteine is the only very stable compound. Although allicin is one the most important biologically active compound found in garlic, it is extremely unstable, and its half-life varies depending on the concentration and temperature of the storage solvent. Additionally, S-allyl-L-cysteine was selected for the NMR quantification due to the presence of an isolate doublet at 5.81 ppm which made it possible to quantify the compound with the applied technique. Thus, S-allyl-L-cysteine was quantified in the extracts of Vessalico garlic, and the French parent cultivars, Messidor and Messidrôme. Results showed that the content of S-allyl-L-cysteine was quite negligible in the two parent cultivars, and this fact allowed us to argue that this compound could be chosen as representative of Vessalico garlic compared to the two French cultivars. The text has thus been modified accordingly (lines 241-246, 338-340, and 566-567).
Because of all this, I cannot recommend this article for publication unless the article is rewritten adding more data to prove that there are indeed metabolites that differentiate this particular variety.
Reviewer 2 Report
Comments and Suggestions for Authors
The paper of Valeria Iobbi et al. “NMR metabolite profiling for the characterization of Vessalico …” aimed to study Vessalico garlic populations by NMR/ SOMs-based approach and determination of antibacterial potential. Generally, paper is well written and includes new scientific information.
Highlights and strengths of the manuscript are:
The results may further increase interest in NMR/ SOMs-based in analysis of natural products and develop new strategies for differentiation of garlic.
Specific comments and suggested revisions:
- Despite your demonstration of the potential value of using Self-Organizing Maps (SOMs) to identify Vessalico garlic, your work used only a method based on the analysis of chemical composition without any application of genetic or other information. It is impossible to claim that some farm is supplying the material of non-Vessalico quality without using a reference method that can validate your conclusions. Apparently, you do not use any method of comparison at all to confirm your conclusions, which are mostly based on speculation on mathematical data. I understand that statistical methods are now increasingly being used to analyze big data and facilitate theoretical work on problems like yours. But even in this case, they are considered only as one of the elements of conjectural conclusions. The authors have done a good job of studying the capabilities of the SOMs for solving the problem of Vessalico garlic identifying but in conclusion it is necessary to indicate the probable shortcomings of the results obtained by the method you used. Otherwise, it seems that the results obtained are absolutely unmistakably correct, although this is not the case.
- It is unclear what the purpose of section 2.3 was. Why S-allyl-L-cysteine was chosen as a marker compound and why only one accessory was analyzed.
Author Response
The paper of Valeria Iobbi et al. “NMR metabolite profiling for the characterization of Vessalico …” aimed to study Vessalico garlic populations by NMR/ SOMs-based approach and determination of antibacterial potential. Generally, paper is well written and includes new scientific information.
Highlights and strengths of the manuscript are:
The results may further increase interest in NMR/ SOMs-based in analysis of natural products and develop new strategies for differentiation of garlic.
Specific comments and suggested revisions:
- Despite your demonstration of the potential value of using Self-Organizing Maps (SOMs) to identify Vessalico garlic, your work used only a method based on the analysis of chemical composition without any application of genetic or other information. It is impossible to claim that some farm is supplying the material of non-Vessalico quality without using a reference method that can validate your conclusions. Apparently, you do not use any method of comparison at all to confirm your conclusions, which are mostly based on speculation on mathematical data. I understand that statistical methods are now increasingly being used to analyze big data and facilitate theoretical work on problems like yours. But even in this case, they are considered only as one of the elements of conjectural conclusions. The authors have done a good job of studying the capabilities of the SOMs for solving the problem of Vessalico garlic identifying but in conclusion it is necessary to indicate the probable shortcomings of the results obtained by the method you used. Otherwise, it seems that the results obtained are absolutely unmistakably correct, although this is not the case.
We thank the reviewer for the insightful comments. We appreciate the opportunity to address reviewer’s concerns and provide further clarification on our approach. However, we agree with the reviewer that our approach presents acknowledges the limitations and potential uncertainties associated with statistical approaches, and we recognize the need for a balanced interpretation of the results, considering SOMs and other multivariate approaches as propaedeutic to further studies. We strive to convey the nuanced nature of our findings and the potential shortcomings inherent in any analytical method. Following the reviewer suggestions, the discussion has thus been modified accordingly (Lines 275, 341-345)
- It is unclear what the purpose of section 2.3 was. Why S-allyl-L-cysteine was chosen as a marker compound and why only one accessory was analyzed.
We thank the reviewer for the comment, and we apologize if the section was not clear. S-allyl-L-cysteine was chosen as potential marker because among the 3 sulfur compounds found as the most discriminant variables, it is the only very stable compound, making it a well-established choice for our analysis. Additionally, S-allyl-L-cysteine was selected for the NMR quantification due to the presence of an isolate doublet at 5.81 ppm which made it possible to easily quantify the compound with the applied technique. Thus, S-allyl-L-cysteine was selected as representative metabolite. The content of S-allyl-L-cysteine was investigated both in Vessalico garlic extract (accession 12) and French parent cultivars (Messidor and Messidrôme). Accession 12 was selected as the most representative among the Vessalico accessions, as resulted from SOM analysis and then confirmed by OPLS-DA. Following the reviewer suggestions, section 2.3 and the discussion have thus been modified accordingly (Lines 241-246 and 338-340)
Reviewer 3 Report
Comments and Suggestions for Authors
The manuscript described a study on a specific ecotype garlic “Vessalico” using the tools of NMR metabolomics. In addition, the extract and individual constituents of the garlic ecotype were tested on Xanthomonas campestris pv. campestris strain. The research was well-designed and executed.
The authors should consider the following remarks:
· In my opinion, the “Introduction” is long and voluminous. It is difficult for the reader to catch the goals of the research. It would be better if the authors managed to optimize it.
· The Figure 1 should be optimized. Use a better resolution and re-order the images - row 1: original image; row 2: A and B; row 3: C and D.
· In Figure 8: Mark the zoomed region in A and B. Use a bigger image for C. Improve the resolution.
· In Table 1: Remove the redundant concentration. In my opinion, only micromolar concentration should be left.
Author Response
The manuscript described a study on a specific ecotype garlic “Vessalico” using the tools of NMR metabolomics. In addition, the extract and individual constituents of the garlic ecotype were tested on Xanthomonas campestris pv. campestris strain. The research was well-designed and executed.
The authors should consider the following remarks:
In my opinion, the “Introduction” is long and voluminous. It is difficult for the reader to catch the goals of the research. It would be better if the authors managed to optimize it.
We thank the reviewer for the suggestion. The introduction has been modified accordingly.
The Figure 1 should be optimized. Use a better resolution and re-order the images - row 1: original image; row 2: A and B; row 3: C and D.
We thank the reviewer for the suggestion. Figure 1 has been changed accordingly.
In Figure 8: Mark the zoomed region in A and B. Use a bigger image for C. Improve the resolution.
We thank the reviewer for the suggestion. Figure 8 has been changed accordingly.
In Table 1: Remove the redundant concentration. In my opinion, only micromolar concentration should be left.
We thank the reviewer for the advice that would make the table simpler from a visual point of view. We considered it necessary to report both concentrations as it is not possible to report the micromolar concentration of the extract and it would therefore be difficult to compare it with the effect of the pure compounds. On the other hand, the micromolar concentration is usually required in the literature for pure compounds. We therefore decided to report both concentrations in the table to not further overload the text.
Round 2
Reviewer 1 Report
Comments and Suggestions for Authors
In its present form, the article can be accepted for publication. I believe, however, that it is a first approximation and that further experiments are needed in order to prove with certainty that it is a different variety from the παρεντ ones and to find other metabolites that will allow it to be certified as a new variety.
Reviewer 2 Report
Comments and Suggestions for Authors
The authors taking into account the observations made by the reviewer. Considering the explanation provided by the authors the paper after correction may accepted in present from.